# AKI3-Risk Predictors and Scores in Radical Nephrectomy with High Thrombectomy under Extracorporeal Circulation for Renal Cell Carcinoma with Supradiaphragmatic Inferior Vena Cava/Right Atrial Thrombus: A Single-Centre Retrospective Study

**DOI:** 10.3390/medicina59020386

**Published:** 2023-02-16

**Authors:** Anca Drăgan, Ioanel Sinescu

**Affiliations:** 1Department of Cardiovascular Anaesthesiology and Intensive Care, Prof. C. C. Iliescu Emergency Institute for Cardiovascular Diseases, 258 Fundeni Road, 022328 Bucharest, Romania; 2Department of Urological Surgery, Dialysis and Kidney Transplantation, Fundeni Clinical Institute, 258 Fundeni Road, 022328 Bucharest, Romania; 3Department of Uronephrology, “Carol Davila” University of Medicine and Pharmacy, 8 Eroii Sanitari Blvd, 050474 Bucharest, Romania

**Keywords:** acute kidney injury, renal cell carcinoma with supradiaphragmatic venous thrombus, extracorporeal circulation, risk scores, intraoperative arterial hypotension

## Abstract

*Background and Objectives*: The recommended therapeutic management in renal cell carcinoma (RCC) with supradiaphragmatic inferior vena cava/right atrial thrombus (IVC/RA) is surgery. Extracorporeal circulation is required. Acute kidney injury (AKI), a frequent complication after nephrectomy and cardiac surgery is associated with long-term kidney disease. This study aims to identify the risk factors involved in the occurrence of the severe postoperative AKI (AKI3) and to analyse various preoperative validated risk scores from cardiac and noncardiac surgery in predicting this endpoint. *Materials and Methods*: The medical data of all patients with RCC with supradiaphragmatic IVC/RA thrombus who underwent radical nephrectomy with high thrombectomy, using extracorporeal circulation, between 2004–2018 in the Prof. C. C. Iliescu Emergency Institute for Cardiovascular Diseases, Bucharest, were retrospectively analysed. The patients who died intraoperatively were excluded from the study. The predefined study endpoint was the postoperative AKI3. Preoperative, intraoperative and postoperative data were collected according to the stratification of study population in two subgroups: AKI3-present and AKI3- absent patients. EuroSCORE, EuroSCORE II, Logistic EuroSCORE, NSQIP any-complications and NSQIP serious-complications were analysed. *Results*: We reviewed 30 patients who underwent this complex surgery between 2004–2018 in our institute. Two patients died intraoperatively. Nine patients (32.14%) presented postoperative AKI3. Age (OR 1.151, CI 95%: 1.009–1.312), preoperative creatinine clearance (OR 1.066, CI 95%: 1.010–1.123) and intraoperative arterial hypotension (OR 13.125, CI 95%: 1.924–89.515) were risk factors for AKI3 (univariable analysis). Intraoperative arterial hypotension emerged as the only independent risk factor in multivariable analysis (OR 11.66, CI 95%: 1.400–97.190). Logistic EuroSCORE (ROC analysis: AUC = 0.813, *p* = 0.008, CI 95%: 0.633–0.993) best predicted the endpoint. *Conclusions*: An integrated team effort is essential to avoid intraoperative arterial hypotension, the only independent risk factor of AKI3 in this highly complex surgery. Some risk scores can predict this complication. Further studies are needed.

## 1. Introduction

Although renal cell carcinoma (RCC) represents around 3% of all cancers [1,2], its incidence rises globally with the highest rates in developed countries [3]. North America, Australia, New Zealand and Europe are the regions with the highest incidence of RCC, while Asia and Africa present the lowest rate [3,4]. RCC has a unique feature: the development of a thrombus in the venous system, via the renal vein to the inferior vena cava (IVC), sometimes even to the right atrium (RA) in 1% of cases [2,5]. Surgery is the only therapeutic solution, regardless of the extent of the thrombus [2]. Otherwise, the prognostic is poor with a median survival period of five months [6]. In RCC with venous extension no surgical method was found superior to another [2,7]. The extent of thrombus was strongly associated with disease-specific mortality in the absence of surgery [6]. Comparison of groups with and without cardiopulmonary by-pass showed no differences in oncological outcomes between the groups [2,7].

In RCC with supradiaphragmatic IVC/RA venous thrombus, the surgery consists of radical nephrectomy with thrombus removal, often under extracorporeal circulation, with or without cross-clamping the ascending aorta arresting the heart. Hypothermia or IVC reconstruction may be sometimes required.

This type of surgery is highly complex. Urologists, cardiac surgeons, cardiologists, anaesthesiologists, perfusionists and many others work together to ensure the perioperative care of these patients. The intraoperative complications can be very serious: massive blood losses with significant transfusion and major embolic events. These can lead to postoperative organ dysfunction or death. One frequent postoperative complication is acute kidney injury (AKI). The radical nephrectomy alone presents a high incidence of AKI, of 53.9% [8], which has been demonstrated to be associated with new onset chronic kidney disease in a graded manner [8]. The usage of extracorporeal circulation for removal of high venous thrombus brings an additional risk for this complication [9]. Previous studies have shown that AKI is associated with long-term morbidity and mortality [10], especially with long-term kidney disease [8,9,10] in major surgery.

That is why it is vital to find the risk factors that influence the occurrence of the most severe stage of AKI (AKI3), especially the modifiable ones, in order to avoid the chronic dialysis requirement. Preoperative risk assessment is also very important [9] in this setting.

In this study, we retrospectively analysed all consecutive patients who underwent radical nephrectomy with supradiaphragmatic IVC/RA thrombectomy using extracorporeal circulation between 2004–2018 in the Prof. C. C. Iliescu Emergency Institute for Cardiovascular Diseases, Bucharest, Romania. We have already shown that intraoperative arterial hypotension was associated with AKI3 in this setting [11]. The present study aims to identify all the preoperative, intraoperative or postoperative risk factors for AKI3 and to assess the role of the already validated risk scores (variants of EuroSCORE and NSQIP) to predict the same endpoint.

## 2. Materials and Methods

This single-centre retrospective study reviewed the medical data of all consecutive patients who underwent radical nephrectomy with supradiaphragmatic IVC/RA thrombectomy using extracorporeal circulation between 2004–2018 in the Prof. C. C. Iliescu Emergency Institute for Cardiovascular Diseases, Bucharest, Romania. The exclusion criteria was intraoperative death. The approval of the Research Ethics Committee (REC) of our institute (12425/15.04.2019) was obtained to access the data and to use it in this study. The requirement for written informed consent was waived by the REC. The predefined study endpoint was the postoperative AKI3. Kidney Disease Improving Global Outcomes (KNIGO) criteria of AKI definition [12] and staging [12] was used to retrospectively evaluate the postoperative renal function, in the surviving patients, using both creatinine and diuresis variation. Preoperative, intraoperative and postoperative data were collected according to the a priori stratification of study population in two subgroups: AKI3-present and AKI3-absent patients.

All the patients were diagnosticated with nonmetastatic RCC with supradiaphragmatic venous thrombus by the urologists. The surgery consists of radical nephrectomy (abdominal time) and high thrombectomy (cardiac time) using extracorporeal circulation, in a multidisciplinary approach, under general anaesthesia, with standard and invasive monitoring (invasive arterial pressure, transoesophageal echocardiography, central venous pressure, diuresis). The postoperative care of the patients was taken in the Department of Cardiovascular Anaesthesiology and Intensive Care of our Institute.

The studied preoperative variables were: age, sex, extent of the thrombus, preoperative creatinine clearance, body mass index (BMI), preoperative nutrition status, preoperative haemoglobin concentration, obesity, kidney involved, the presence/absence of other comorbidities. The extent of the thrombus was determined using transoesophageal echocardiography. The preoperative creatinine clearance was evaluated using the last preoperative value of serum creatinine in the Cockcroft–Gault Equation. We used the last preoperative serum creatinine value and haemoglobin concentration in our analysis. BMI classified the patients according to their nutrition status. The presence/absence of preoperative diabetes mellitus, arterial hypertension and heart failure were identified according to the diagnosis recorded in the preoperative medical evaluation.

The studied intraoperative variables were: duration of extracorporeal circulation, intraoperative blood loss, minimum intraoperative haemoglobin concentration, units of packed red blood cell transfused intraoperatively, units of fresh frozen plasma transfused intraoperatively, usage of cross-clamping of ascending aorta, duration of cross-clamping of ascending aorta, using of ultrafiltration during the extracorporeal circulation, temperature management during extracorporeal circulation (moderate hypothermia/normothermia) and intraoperative arterial hypotension. The presence of the intraoperative arterial hypotension was defined as the mean intraoperative arterial blood pressure (MAP) below 60 mmHg for more than 20 min before, during and after extracorporeal circulation. All the patients had continuously monitored invasive arterial pressure via an arterial catheter such that the anaesthesiologist could document MAP in the anaesthesia sheet at 5–15 min intervals.

The postoperative variables were: haemostasis reintervention and systemic inflammatory response syndrome (SIRS). SIRS was retrospectively evaluated according to the postoperative medical data, using the recognized definition [13].

We reviewed EuroSCORE, EuroSCORE II, Logistic EuroSCORE, NSQIP any-complications and NSQIP serious-complications for the two subgroups of patients and expressed them as quantitative variables. We have chosen EuroScore, Logistic EuroScore and EuroScore II to be assessed to our endpoint because our surgery consists of a cardiac time (with extracorporeal circulation) beside the abdominal time (radical nephrectomy).

We used IBM SPSS Statistics 26 in our analysis with a threshold of statistical significance of 95% (*p* < 0.05). The quantitative variables were displayed with their mean/median and standard deviation (SD) and analysed using the independent *t*-test or Mann–Whitney test depending on the variable distribution (the Shapiro-Wilk test). The categorical variables were analysed using the exact Fisher test (for 2 × 2 contingency tables) or Kendall tau b test (if chi test cannot be applied). Receiver operator characteristic (ROC) analysis was used to assess the quantitative variables to the endpoint (AUC, *p*, CI 95%). The cut-offs with their sensitivity and specificity were presented. ROC analysis was also used to asses and to compare the risk scores. The statistically significant variables have been tested in univariable binary logistic regression (Exp(B), OR, CI 95%, *p*). After testing for multicollinearity, the statistically significant variables were introduced in the multivariable binary logistic regression to identify the independent risk factors for AKI3 (OR, CI 95%, *p*).

## 3. Results

We reviewed 30 consecutive patients who underwent radical nephrectomy with supradiaphragmatic IVC/RA thrombectomy using extracorporeal circulation between 2004–2018 in the Prof. C. C. Iliescu Emergency Institute for Cardiovascular Diseases, Bucharest, Romania. Two patients who died intraoperatively were excluded from the study. Figure 1 presents the diagram of the studied patients. Nine patients (32.14%) presented postoperative AKI3 (AKI3-present).

Table 1 displays the preoperative, intraoperative and postoperative variables for the two subgroups of patients.

Sex, nutritional preoperative status, obesity, preoperative diabetes mellitus, arterial hypertension or heart failure, performing left/right radical nephrectomy or the extent of thrombus were not associated with postoperative AKI3 (Table 1). The preoperative haemoglobin concentration, the minimum intraoperative haemoglobin concentration, the intraoperative blood loss or the transfusion needed intraoperatively were not significantly different in the two subgroups of patients (Table 1).

The nine AKI3-present patients were significantly older (63.33 ± 7.01 years) than the others (54.42 ± 9.83 years old) (Table 1). The ROC analysis showed (Figure 2) that patient’s age was significantly associated with the postoperative AKI3 with an AUC of 0.769 (*p* 0.024, CI 95%: 0.584–9.954). A cut off value of 60.5 years predicted the endpoint (sensitivity 66.7%, specificity 73.7%).

The preoperative creatinine clearance was significantly lower (59.44 ± 21.57 mL/min versus 82.10 ± 17.83 mL/min) in AKI3-present patients (Table 1) as shown by the ROC analysis with a significant AUC (Figure 3) of 0.825 (*p* 0.006, CI95%: 0.630–0.954). A cut-off value of 62.50 mL/min predicted the outcome (sensitivity 94.7%, specificity 77.80%).

The intraoperative arterial hypotension was significantly associated with the postoperative AKI3 (Table 1). The cross-clamping of the ascending aorta was used only in 20 patients (71.42%). Seven (35%) of these patients developed AKI3. The duration of aorta cross-clamping was longer in AKI3-present (51.83 ± 14.47 min) compared to AKI3-absent patients (36.54 ± 34 min), but the result had no statistical significance (*p* = 0.311). Using cross-clamping of ascending aorta, the duration of extracorporeal circulation, ultrafiltration or different temperature management on extracorporeal circulation did not influence the endpoint (*p* > 0.05, Table 1). Haemostasis reinterventions or postoperative SIRS have not been associated with postoperative AKI3 (*p* > 0.05, Table 1).

Intraoperative arterial hypotension and the two variables with a significant AUC in predicting the endpoint, age and preoperative creatinine clearance, also presented statistically significant results in univariable binary logistic regression (*p* ˂ 0.05 in Table 2). Increasing age (Exp(B) = OR = 1.151, CI 95% 1.009–1.312) and the presence of intraoperative arterial hypotension (Exp(B) = OR = 13.125, CI 95% 1.924–89.515) are risk factors of AKI3 in univariate analysis. The lower preoperative creatinine clearance is risk factor in univariable binary logistic regression for postoperative AKI3 (Exp(B) = 0.938, OR = 1.066 with CI 95% 1.010–1.123). These 3 variables were introduced in the multivariable logistic regression after checking for multicollinearity. The intraoperative arterial hypotension was the only variable that emerged independently associated with AKI3 in multivariable analysis (OR 11.66, CI 95%:1.400–97.190). Age and preoperative creatinine clearance did not present significant results in this analysis (*p* > 0.05 in multivariable binary logistic regression in Table 2).

EuroSCORE, EuroSCORE II, Logistic EuroSCORE, NSQIP any-complications and NSQIP severe-complications were assessed for the patients in the two subgroups. Their values were significantly higher in the AKI present-patients and their values were significantly different, except for NSQIP any-complications (Table 3).

We have compared the studied risk score toward the occurence of postoperative AKI3 using the characteristics of their ROC curves (Table 4).

The Logistic EuroSCORE and EuroSCORE (additive) predicted the endpoint with good discrimination (AUC ˃ 0.8, Table 4). The cut off value of 2.43 of the Logistic EuroSCORE in relation with AKI3 with a sensitivity of 88.9% and a specificity of 68.4%. EuroSCORE presented a cut off of 3.5 in assessing our endpoint, with a sensitivity of 77.8% and a specificity of 68.4% (Table 4).

EuroSCORE II and NSQIP severe-complications also had statistically significant fair values of areas under the ROC curve (AUC between 0.7 and 0.8), but NSQIP any-complications did not present significant AUC, also its value is more than 0.7 (Table 4).

## 4. Discussion

Age, preoperative clearance creatinine and intraoperative arterial hypotension were the risk factors for postoperative AKI3 in our univariable analysis. But intraoperative arterial hypotension emerged as the only independent risk factor. This was defined as MAP under 60 mmHg for more than 20 min pre, during and post extracorporeal circulation The result emphasizes the previous findings of our group that demonstrated a significant association between intraoperative arterial hypotension and the postoperative complications, including AKI3 [11]. Our present study identified the risk factors for AKI3 in this complex surgery using preoperative, intraoperative and postoperative variables. We showed that Logistic EuroSCORE best predicted the outcome, followed by EuroSCORE, EuroSCORE II and NSQIP serious-complications.

The patients included in our study underwent radical nephrectomy with supradiaphragmatic IVC/RA thrombectomy using extracorporeal circulation. Both radical nephrectomy and cardiac surgery present high risk for AKI, but our studied surgery is a combination of them in oncological patients.

AKI definition and staging were elaborated by KNIGO [12]. Some factors are strongly related to postoperative AKI: cardiac surgery, especially with cardiopulmonary bypass, major noncardiac surgery, sepsis, circulatory shock, aged patients, women, anemia, diabetes mellitus, chronic heart failure etc [12]. Other studies also reported that postoperative AKI depends on the type of surgery [14]. Cardiac, thoracic, orthopedic, vascular and urologic surgery have an increased AKI incidence [14]. Minimal changes in postoperative renal function can lead to high mortality and morbidity [14]. A meta-analysis demonstrated a global incidence of AKI in cardiac surgery of 25–30% [15,16], 13.6% for AKI 1, 3.8% for AKI 2, 2.3% for AKI3 [15]. Others showed that severe AKI incidence is 0.33% to 9.5% in cardiac surgery [17]. Age, preoperative creatinine, duration of extracorporeal circulation, diabetes mellitus, heart failure and feminine sex are risk factors for AKI that need Continuous Renal Replacement Therapy [18]. In cardiac surgery, the AKI-patients have a cumulative death risk at 5 years of 26%, more than double compared to no-AKI patients [19]. The preoperative hypertension, heart failure, diabetes mellitus, creatinine, age, high BMI, the duration of cross-clamping of ascending aorta, low cardiac output and prolonged arterial hypotension are AKI independent risk factors in Gumbert’s review [20]. Intraperitoneal surgery, hemodynamic instability, intraoperative transfusion are risk determinants in noncardiac surgery [20].

Radical nephrectomies had a high AKI risk [8]. Male sex, older age [21,22], radical nephrectomy [21], comorbidities, preoperative chronic renal disease [21], high BMI [22] are AKI risk factors in urology. Chronic kidney disease was more frequent in AKI-patients at 1 year (54.7% versus 43.9%) and 3 years (50% versus 32%) [22]. The postoperative AKI and long-time renal function deterioration were associated in a progressive manner [8]. The highest incidence of 53.9% was reported in patients with radical nephrectomy and simultaneous IVC thrombectomy [20]. A retrospective study showed that postoperative AKI in RCC-patients can lead to chronic renal failure, prolonged hospitalization and an increased rate of mortality at one year [23]. But only ten patients (13.1%) needed extracorporeal circulation in their study. The comorbidities, the extent of thrombus, preoperative creatinine and haemoglobin concentration, intraoperative blood loss, number of red packed units did not correlate with postoperative AKI [23]. Male sex and prolonged duration of IVC cross-clamping were strongly related to AKI [23].

The high incidence of the AKI in radical nephrectomy would be due to the loss of nephrons in the resected kidney, an unmodifiable factor, but also to damage of the other kidney. It is already known that in donor nephrectomy compensatory hypertrophy develops in the contralateral kidney and, thus, the kidney function remains almost normal [24,25]. The patients studied by our group have been diagnosticated with renal cell carcinoma and present a lot of comorbidities. Their perioperative management is challenging. Preoperative assessment and optimisation are mandatory to be done. Avoiding nephrotoxic medication especially nonsteroidal anti-inflammatory drugs and angiotensin converting enzyme inhibitors, maintaining adequate volume status, using a responsible management of the contrast in preoperative imagistic evaluation may be some actions [12,26].

Intraoperatively, maintaining the optimum renal perfusion of the remaining kidney seems to be the most important action in preventing AKI and especially AKI3. The multidisciplinary team must avoid the hemodynamic instability and manage the low cardiac output states, rationally using the vasopressors and inotropes. Hypovolemia, anaemia and vasoplegia must be also addressed in order to optimise the arterial renal perfusion of the remaining kidney [26]. The increased renal after load must be also avoided using appropriate lung ventilation with optimised peep, diuresis, ultrafiltration [26].

The diabetes mellitus, arterial hypertension or heart failure, preoperative haemoglobin concentration, the extend of thrombus, sex, kidney involved or preoperative nutritional status did not associate in our study with AKI3. Intraoperative blood loss, the level of required transfusion, the duration of extracorporeal circulation, minimum haemoglobin concentration, use of ultrafiltration, temperature management, hemostatic reintervention or SIRS have the same characteristics. Using the cross-clamping of the aorta and its duration did not influence AKI3 in our study.

Considering other studies results, it is not a surprise that in our study intraoperative arterial hypotension, age and preoperative creatinine clearance were significantly correlated with AKI3. Thus, we emphasize the importance of the close invasive monitoring of the arterial pressure and of the multidisciplinary as well to minimize the magnitude and the duration of the intraoperative arterial pressure, but also to optimise preoperatively the renal function of the patients.

In noncardiac surgeries, the maintenance of sufficient MAP is the most important hemodynamic parameter to preserve perioperatively [9]. The 2019 perioperative consensus stated that a MAP under 60–70 mmHg is associated with myocardial injury, AKI and death, the effect being a function of magnitude and duration of intraoperative hypotension [27]. Targeting sufficient MAP is also proposed by the guidelines regarding cardiopulmonary bypass [28]. Thus, MAP is accepted in a range of 50–80 mmHg [28]. Targeting a higher blood pressure during extracorporeal circulation does not reduced AKI [29]. Today, optimal MAP during nonpulsatile cardiopulmonary bypass is not known yet [9,22,28]. This is proposed to be assessed using novel approaches, further evidence being required [9,28].

Since 1999, EuroSCORE, has been demonstrated to be a tool for predicting mortality in cardiac surgery [30,31]. Using the logistic regression equation this risk score was transformed in Logistic EuroSCORE. In 2003, this model was reported to be a better risk predictor, especially for high risk patients, although it is more difficult to be performed in practice, requiring computational tools [32]. It was proved that additive EuroSCORE overestimates mortality, but the mortality of the high-risk patients in cardiac surgery was shown to be underestimated by this risk score [33]. Niv Ad et al. in 2016 demonstrated that EuroSCORE II provides a better prediction of mortality than the first variant of the score [34]. This is why EuroSCORE II is proposed nowadays as a risk score in complex cardiac surgery [34].

Regarding the risk scores for AKI, Toumpoulis et al. raised the question since 2004 if EuroSCORE can predict anything else than mortality [35]. They showed that EuroSCORE and Logistic EuroSCORE could predict postoperative acute renal failure in cardiac surgery [29]. Studying 4651 patients undergoing cardiac surgery, Duthie et al. also demonstrated that EuroSCORE could identify high-risk patients for postoperative AKI [36]. Chen et al. study reported that EuroSCORE and EuroSCORE II successfully predicted AKI3 in aortocoronary bypass surgery [37]. 

NSQIP was proposed to predict the risk of postoperative complications. Both NSQIP serious and any-complications refer to progressive renal insufficiency and acute renal failure [38]. Some studies doubt about the capacity of NSQIP of predicting mild and moderate acute renal injuries [39], especially in urologic surgery. Frazier et al. in 2014 reported that that although urology has been among the surgical specialities introduced in this system ever since it was founded, there are poor date regarding urology [40]. Blair et al. made a comparison in 2018 between the predicted results and those observed 30 days after surgery in patients with partial nephrectomy for renal carcinoma, concluding that NSQIP underestimates postoperative complications [41]. Anyhow, American Urological Association recommend using NSQIP together with other risk assessment modalities [42].

In our study, in nephrectomy with high thrombectomy using extracorporeal circulation, all the variants of the cardiac surgery risk scores predicted AKI3. Logistic EuroSCORE, followed by EuroSCORE had the highest predictive value (AUC > 0.800). EuroSCORE II and NSQIP serious-complications also predicted AKI3, but with AUC < 0.800. That is why we propose the three variants of EuroSCORE, especially Logistic EuroSCORE and EuroSCORE, as a tool to predict AKI3 in this kind of surgery. EuroSCORE may be even easier to use, comparing to its logistic formula. We must take into consideration the calculated cut off of EuroSCORE of 3.5. This mean that an EuroSCORE more or equal to four, not a high value, highlights the risk of postoperative AKI3. NSQIP serious-complications may be also used, but probably together with another risk score. To the best of our knowledge, this is the first study to assess these risk scores in predicting AKI3 in this complex surgery.

Study limitation. This study is a single-centre retrospective study, with a relatively small number of patients, due to the reduced incidence of this pathology, renal cell carcinoma with supradiaphragmatic venous extension. Our Institute, the Prof. C. C. Iliescu Emergency Institute for Cardiovascular Diseases, Bucharest, represents the most experienced hospital in this field in Romania. Almost all the studies regarding this complex surgery are single-centre and retrospective. The most experienced centre worldwide, Cleveland Clinic, reports 101 cases of this kind in a 28 years-experience [43]. Even if the patients’ number is small in our study, its clinical importance is significant because this complex surgery with high complications rate represents the only treatment choice for these patients. We consider that all scientific data in this field can be useful. Further studies are needed to confirm the results.

## 5. Conclusions

The radical nephrectomy with supradiaphragmatic inferior vena cava/right atrial thrombectomy performed using extracorporeal circulation for safe removal of the high thrombus in patients diagnosticated with nonmetastatic renal cell carcinoma is a complex surgery. Acute kidney injury, a frequent complication both in radical nephrectomy and in cardiac surgery, presents a high incidence in this setting. Preoperative risk assessment and renal function optimisation are useful, as we demonstrated that preoperative creatinine clearance, a partially modifiable risk factor, determined AKI3 in univariable analysis. Some validated risk scores from cardiac surgery can predict this complication, especially Logistic EuroSCORE and EuroSCORE. NSQIP serious-complications may be used to complete preoperative evaluation. Intraoperatively, all the effort must be made to maintain the optimal perfusion pressure of the remaining kidney especially monitoring the invasive blood pressure and avoiding prolonged periods of arterial hypotension. Of course, due to the limitations of our study, further studies are needed to confirm our results.

## Figures and Tables

**Figure 1 medicina-59-00386-f001:**
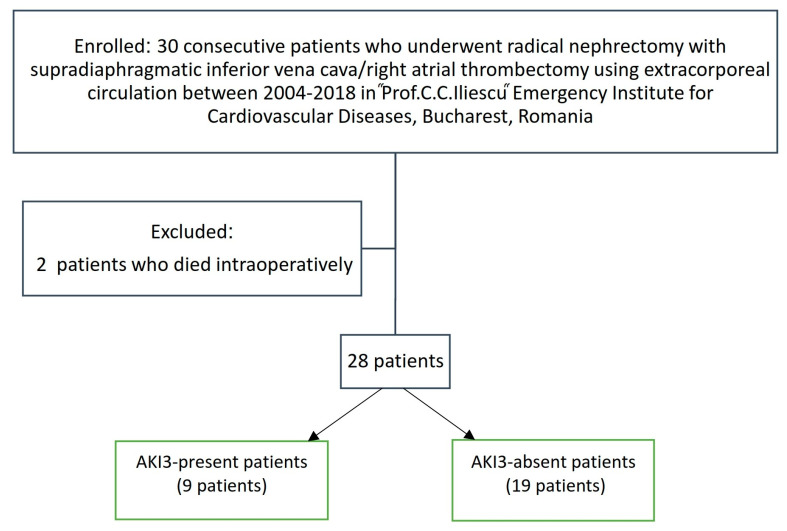
The diagram of the studied patients.

**Figure 2 medicina-59-00386-f002:**
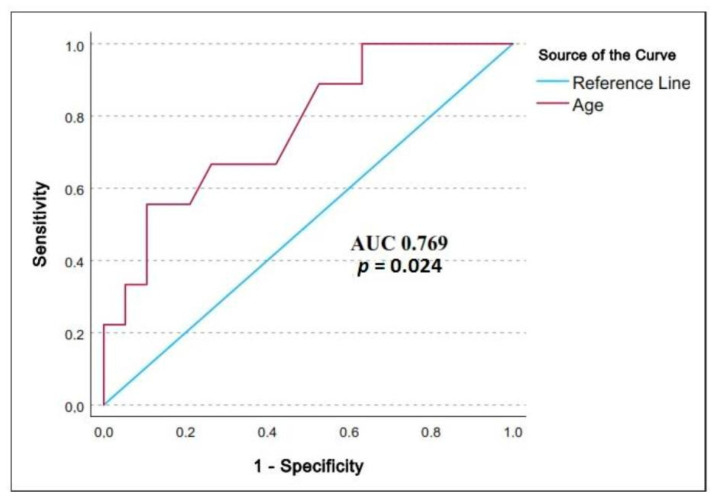
Receiver Operator Characteristic curve analysis for age predicting the study endpoint. We present: AUC, area under curve; *p*, probability value.

**Figure 3 medicina-59-00386-f003:**
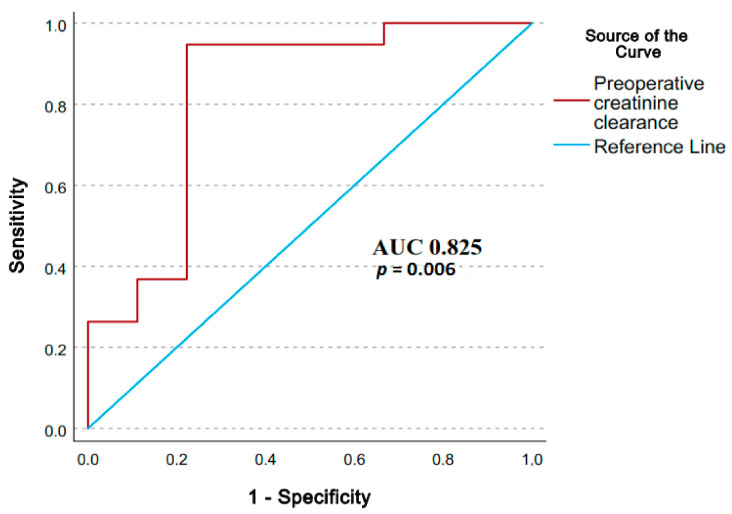
Receiver Operator Characteristic curve for preoperative creatinine clearance predicting the study endpoint. We present: AUC, area under curve; *p*, probability value.

**Table 1 medicina-59-00386-t001:** Characteristics of variables in the two subgroups of patients from the study population (*n* = 28).

Variable	AKI3 - Present Patients (*n* = 9)	AKI3 - Absent Patients (*n* = 19)	*p* ^1^
Age ^2^	63.33 ± 7.01	54.42 ± 9.83	**0.022**
Preop.Cl.creat (mL/min) ^2^	59.44 ± 21.57	82.10 ± 17.83	**0.007**
BMI (kg/m^2^) ^2^	26.94 ± 4.09	26.55 ± 2.74	0.588
Preop.Hb (g/dl) ^2^	11.17 ± 1.96	11.52 ± 2.06	0.681
Duration of EC (min) ^2^	92.20 ± 58.60	61.84 ± 25.15	0.209
Intraoperative blood loss (litres) ^2^	6.92 ± 4.97	5.33 ± 4.39	0.332
Cmin Hb (g/dl) ^2^	7.10 ± 0.78	6.82 ± 1.31	0.523
pRBC ^3^	9 ± 3.90	4 ± 5.18	0.085
FFP ^3^	8 ± 5.41	4 ± 5.58	0.468
Male gender ^4^	4 (44.44%)	15 (78.94%)	0.097
Right kidney involved ^4^	4 (44.44%)	15 (78.94%)	0.097
The extent of thrombus to the right atrium ^4^	6 (66.66%)	10 (52.63%)	0.687
Nutritional status ^4^ N/W/O	5/2/2 (55.55%/22.22%/22.22%)	4/14/1 (21.05%/73.68%/5.26%)	0.446
Obesity ^4^	2 (22.22%)	1 (5.26%)	0.234
Preoperative diabetes mellitus ^4^	2 (22.22%)	2 (10.52%)	0.574
Preoperative arterial hypertension ^4^	8 (88.88%)	13 (68.42%)	0.371
Preoperative heart failure ^4^	6 (66.66%)	8 (42.10%)	0.420
Cross-clamping ascending aorta ^4^	7 (77.77%)	13 (68.42%)	1
Ultrafiltration on EC ^4^	1 (11.11%)	5 (26.31%)	0.630
Temperature management on EC ^4^ H/Normothermia ^4^	6/3 (66.66%, 33.33%)	5/14 (26.31%/73.68%)	0.095
hAP ^4^	7 (77.77%)	4 (21.05%)	**0.010**
Haemostasis reinterventions ^4^	1 (11.11%)	1 (5.26%)	1
SIRS ^4^	7 (77.77%)	8 (42.10%)	0.114

BMI: body mass index; Cmin.Hb: minimum intraoperative haemoglobin concentration; EC: extracorporeal circulation; FFP: units of fresh frozen plasma transfused intraoperatively; H: moderate hypothermia; hAP: intraoperative arterial hypotension; N: normal preoperative nutrition status (BMI 18.5–24.9kg/m^2^); O: obesity (BMI ≥ 30kg/m^2^); Preop.Cl.creat: preoperative creatinine clearance; Preop.Hb: preoperative haemoglobin concentration; *p*: probability value; pRBC: units of packed red blood cell transfused intraoperatively; SD: standard deviation; SIRS: systemic inflammatory response syndrome; W: overweight (BMI 25–29.9 kg/m^2^) ^1^ presents *p* value of independent *t*-test/Mann–Whitney test for quantitative variables and exact Fischer test for categorical variables; significant results are bolded. ^2^ data are presented as mean ± SD. ^3^ data are presented as median ± SD. ^4^ data are presented as n, %.

**Table 2 medicina-59-00386-t002:** The results of univariable and multivariable binary logistic regression.

Variable	Univariable Binary Logistic Regression			Multivariable Binary Logistic Regression	
	Exp(B)	OR (CI 95%)	*p*	OR (CI 95%)	*p*
Age	1.151	1.151 (1.009–1.312)	**0.036**	1.093 ( 0.925–1.291)	0.297
Preop.Cl.creat	0.938	1.066 (1.010–1.123)	**0.020**	1.038 (0.973–1.108)	0.248
hAP	13.125	13.125 (1.924–89.515)	**0.009**	11.66 (1.400–97.190)	**0.038**

Abbreviations: CI: confidence interval; hAP: intraoperative arterial hypotension; Exp(B): exponential value of B; OR: odds ratio; Preop.Cl.creat: preoperative creatinine clearance; *p*: probability value, significant results are bolded.

**Table 3 medicina-59-00386-t003:** Characteristics of the risk scores in the two subgroups of patients.

Risk Score	AKI3 Absent-Patients ^1^ (*n* = 19)	AKI3 Present-Patients ^1^ (*n* = 9)	*p* ^2^
Logistic EuroSCORE	1.61 ± 1.21	4.16 ± 1.75	**0.007**
EuroSCORE	2 ±1.35	5 ± 1.80	**0.008**
EuroSCORE II	0.87 ± 0.81	1.87 ± 0.82	**0.012**
NSQIP severe-complications	21.80 ± 3.31	21.80 ± 3.01	**0.033**
NSQIP any-complications	24 ± 3.89	24.20 ± 3.51	0.064

Abbreviations: AKI3: severe acute kidney injury; NSQIP: National Surgical Quality Improvement Program; *p*: probability value; SD: standard deviation; ^1^ data are presented as median ± SD. ^2^
*p* value of Mann-Whitney test; significant results are bolded.

**Table 4 medicina-59-00386-t004:** Characteristics of the ROC curve of the studied risk scores in predicting the postoperative AKI3.

Risk Score	ROC Curve			Cut Off	Sensitivity (%)	Specificity (%)
	AUC	*p*	CI 95%			
Logistic EuroSCORE	0.813	0.008	0.633–0.993	2.43	88.9	68.4
EuroSCORE	0.801	0.011	0.619–0.983	3.50	77.8	68.4
EuroSCORE II	0.798	0.012	0.590–0.986	1.32	88.9	78.9
NSQIP severe-complications	0.737	0.046	0.553–0.921	21.45	100	57.9
NSQIP any-complications	0.713	0.073	0.520–0.907	-	-	-

AUC: area under curve; CI: confidence interval; NSQIP: National Surgical Quality Improvement Program; ROC:Receiver Operator Characteristic; *p*: probability value.

## Data Availability

Data supporting reported results can be found at the following email address: anca.dragan1978.14@gmail.com.

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
