# Peer review of "AKI3-Risk Predictors and Scores in Radical Nephrectomy with High Thrombectomy under Extracorporeal Circulation for Renal Cell Carcinoma with Supradiaphragmatic Inferior Vena Cava/Right Atrial Thrombus: A Single-Centre Retrospective Study"

_medicina, 2023, doi:10.3390/medicina59020386_

Round 1

Reviewer 1 Report

The paper describes the predictiors of AKI3 in patients who underwent nephrectomy with thrombectomy for supradiaphragmic RCC. The paper is well-written, the methodology is correct, results are clearly presented. But, some changes and explanations are required.

Please see the details in the following points.

1.      “Renal cell carcinoma (RCC), a low incidence neoplasm..” This sentence should be corrected. It is not a rare neoplasm. Please provide more detailed statistics on incidence.

2.      In the methods section you mention “sensibility” instead of “sensitivity”

3.      In the discussion a lot of sentences present once again the results of the study. Some of them (insignificant associations) are not relevant to mention again. Please try to modify the discussion and more robustly comment on factors causing AKI-3 during the surgery including the proposed pathophysiological mechanism and reports of other authors.

4.      Please could You provide some details about the oncological and survival outcomes of patients? Did any of the patients have metastasis at the time of surgery? How long were the hospital stays?

5.      Conclusions of the paper should not recall once again the definition of intraoperative hypotension

Thank You!

Reviewer 2 Report

The study establishes a comparison between the clinical data of 30 patients underwent radical nephrectomy with supradiaphragmatic IVC/RA thrombectomy using extracorporeal circulation.

Although the manuscript is well written and detailed, the study´s results are nothing new to the field. There are previous reports showing the development of AKI after surgery. It is already well established that creatinine clearance as well as blood pressure are indicators of kidney damage.

The results are not surprising, the analysis is modest and simplified. the results do not provide a development of the current knowledge in the field.

Mayors

Authors should explain in detail tables 2 and 3. Please explain to the readers the use of EuroScore and the data showed in this tables.

Authors must Include exclusion criteria in methods

Minors

Figure 3 does not provide necessary or important visual information for the manuscript.

Round 2

Reviewer 1 Report

Thank you

Reviewer 2 Report

The content of the article has improved. The authors should be recognize for adding and modifying information in a way that makes the message clearer for the reader.